# Role of Angiogenesis in Retinal Diseases and New Advances in Drug Development

**DOI:** 10.3390/cells14231849

**Published:** 2025-11-24

**Authors:** Emma Boey, Humza Zaidi, Tina Tang, Amirfarbod Yazdanyar

**Affiliations:** 1Albert Einstein College of Medicine, 1300 Morris Park Ave, Bronx, NY 10461, USA; 2University of Connecticut School of Medicine, 200 Academic Wy, Farmington, CT 06032, USA; 3Department of Ophthalmology and Vision Sciences, University of Toronto, Toronto, ON M5R 0A3, Canada; 4Retina Group of New England, 174 Cross Rd, Waterford, CT 06385, USA

**Keywords:** retinal diseases, neovascular age-related macular degeneration, diabetic retinopathy, retinal vein occlusion, retinopathy of prematurity, angiogenesis inhibitors, anti-angiogenic

## Abstract

Dysregulation of angiogenesis can cause a disruption in oxygen and nutrient delivery, resulting in impaired neural retinal function. Understanding the underlying components involved in its pathophysiology is essential to develop new treatments for preserving and restoring vision. The aim of this review is to describe the role of angiogenesis in different retinal and choroidal pathologies and evaluate current and emerging anti-angiogenic therapies for retinopathies. Current research articles, focusing on the latest clinical trials from the last two decades, were used to write this review. We discuss normal angiogenesis, in contrast to pathological angiogenesis, in four diseases: retinal vein occlusion (RVO), age-related macular degeneration (AMD), diabetic retinopathy (DR), and retinopathy of prematurity (ROP). Alongside these diseases, this review discusses relevant anti-angiogenic therapies that have been approved for use and are under active investigation through clinical trials for their safety and efficacy.

## 1. Introduction

Retinal vasculature development is vital for normal visual function. Angiogenesis, the process of vessel formation from pre-existing vasculature, is controlled by a balance between proangiogenic and anti-angiogenic factors. In the normal retina and choroid, angiogenesis is maintained by a tightly regulated interplay of signaling factors to prevent pathological neovascularization. The retina is supplied by the central retinal artery and choriocapillaris, which are responsible for the inner retina and outer retina, respectively [1].

Between 6 and 8 weeks of gestation, the choriocapillaris develops through hemovasculogenesis, a process that involves the simultaneous development of vascular structures and primitive blood cells [1,2]. The three layers of choroidal vessels are fully developed by 21 weeks of gestation, which coincides with the onset of photoreceptor cell differentiation, underscoring the vascular and neural coordination of retinal development to ensure adequate metabolic support [2]. From 16 to 40 weeks’ gestation, superficial retinal vascular layers develop, starting from the optic nerve head and progressing toward the peripheral edge until they dive into the retina to form the deep retinal vascular layer at the base of the outer plexiform layer (OPL), followed by an intermediate layer between the superficial and deep networks [3].

Within retinal vascular development, oxygen plays an important role. When the metabolic demand for differentiating retinal neurons and glial cells increases, physiological hypoxia triggers the development of new vessels. The effect of oxygen on vascular growth is mediated largely by VEGF, a hypoxia-induced growth factor that is expressed in the developing retina. CXC chemokine receptor (CXCR)-4 and its ligand SDF-1 are also hypoxia-induced and are highly expressed in the innermost retina (Table 1). Erythropoietin is another hypoxia-induced growth factor that also has neuroprotective and pro-angiogenic effects. Hypoxia-regulated growth factors are controlled by hypoxia-inducible factor (HIF) [1,4]. Non-oxygen-regulated growth factors include Tie2 receptors, Tie2 ligand angiopoietin 2 (Ang-2), and insulin-like growth factor-1 (IGF-1) (Table 1) [5,6].

Dysregulation of angiogenesis can disrupt the delivery of oxygen and nutrients, resulting in the disruption of neural retinal function. Understanding the underlying components involved in its pathophysiology is essential to develop new treatments to preserve and restore vision. Neovascularization (NV) arising from retinal circulation is commonly observed in retinal diseases, including diabetic retinopathy (DR), retinal vein occlusion (RVO), and retinopathy of prematurity (ROP). NV arising from the choroidal circulation is seen in neovascular age-related macular degeneration (nAMD). Persistent inflammation in these diseases promotes NV and fibrotic remodeling through the upregulation of pro-angiogenic and pro-permeability cytokines (PLGF, PDGF, IL-6, IL-8, MCP-1, ICAM-1, IP-10, and EPO), breakdown of the blood-retinal barrier (BRB), and chronic tissue injury [7,8,9]. Furthermore, activation of the complement system, particularly through polymorphisms in CFH, C3, C2/BF, has been shown to lead to chronic low-grade inflammation and recruitment of macrophages and microglia to the subretinal space, particularly in AMD (Table 1). These immune cells then secondarily release pro-angiogenic cytokines, including interleukin-6 (IL-6), tumor necrosis factor-alpha (TNF-α), and monocyte chemoattractant protein-1 (MCP-1), which further enhance VEGF expression and neovascularization [10,11]. This review aims to describe the role of angiogenesis in different retinal and choroidal pathologies and evaluate current and emerging anti-angiogenic therapies for retinopathies.

## 2. Retinal Vein Occlusion

### 2.1. Retinal Vein Occlusion—Classifications and Pathogenesis

Retinal vein occlusion (RVO) is classified into branch retinal vein occlusion (BRVO) and central retinal vein occlusion (CRVO) based on the location of venous blockage. CRVO is further subdivided into ischemic CRVO (iCRVO) and non-ischemic CRVO. Hemi-retinal vein occlusion (HRVO) is a clinical intermediate of BRVO and CRVO [12]. In CRVO, arteriosclerotic and hyperplastic changes to the thick central retinal artery cause lumen narrowing of its adjacent thin-walled central retinal vein. BRVO involves the compression of branch retinal veins by crossing rigid arteries, leading to subsequent turbulent blood flow and injury to the venous endothelium [13]. This leads to hypoxia and HIF-1 activation, which induces the central mediator VEGF. The binding of Ang-2, which is activated in hypoxia, is also implicated in RVO pathogenesis. The simultaneous activation of VEGF and Ang-2/Tie2, along with the activation of pro-inflammatory cytokines, works to increase vascular permeability, break down the BRB, and further perpetuate fluid leakage and macular edema [14].

### 2.2. Retinal Vein Occlusion—Classifications and Pathogenesis- Current Anti-Angiogenic Therapies

Anti-VEGF agents play a pivotal role in the treatment of all subtypes of RVO. Three anti-VEGF agents are currently FDA-approved for RVO: ranibizumab, aflibercept, and faricimab. Bevacizumab is commonly used as an off-label treatment. Bevacizumab is a humanized full-length antibody that targets all soluble isoforms of VEGF-A. Although it is not FDA-approved as a treatment for RVO, it has been studied for its efficacy in reducing macular edema and increasing visual gain for both CRVO and BRVO [15]. The results of the SCORE2 clinical trial showed no significant difference between bevacizumab and aflibercept in terms of visual acuity over 6 months of follow-up in patients treated with CRVO or HRVO [16]. Other studies have investigated the efficacy and frequency of intravitreal aflibercept versus bevacizumab in CRVO and have found comparable effectiveness for the two treatments without significant complications; however, the frequency of injections in the aflibercept group was lower [17,18]. Similarly, studies comparing the efficacy of bevacizumab with ranibizumab in patients with both subtypes of RVO found that bevacizumab was non-inferior to ranibizumab for patients with macular edema receiving monthly injections for a period of 6 months [19]. Notably, the post hoc analysis of the LEAVO trial found that, despite its cost-effectiveness, bevacizumab was non-inferior to ranibizumab and aflibercept in the intention-to-treat population; however, bevacizumab was not durable and required more injections in patients with macular edema due to CRVO [20].

Ranibizumab is a recombinant, humanized Fab antibody fragment that recognizes all isoforms of VEGF-A. Observational studies have shown comparable visual outcomes and similar number of injections versus aflibercept in the management of macular edema secondary to CRVO [21,22,23,24,25,26]. Beyond visual acuity, ranibizumab has been shown to reduce retinal hemorrhage, the development of disc neovascularization, and papilledema [27]. 24-month results from the CRYSTAL clinical trial found that best-corrected visual acuity (BCVA) gains after at least 3 months of consecutive monthly 0.5 mg ranibizumab were sustained in patients with visual impairment due to macular edema secondary to CRVO [28]. The BLOSSOM and LUMINOUS studies reported improvements in visual acuity among a geographically diverse set of patients with BRVO [29,30].

Human fusion proteins are newer anti-VEGF agents that are being used for macular edema secondary to RVO. Aflibercept is a humanized recombinant fusion protein that neutralizes VEGF-A, VEGF-B, and PLGF (both PLGF-1 and -2). Studies comparing the efficacy of intravitreal injections of ranibizumab with that of aflibercept have shown no significant improvement in BCVA between the two medications, with varying findings on the necessary injection frequencies among studies and anatomical improvements in patients with macular edema secondary to BRVO and CRVO [23,25,31]. In studies comparing intravitreal aflibercept with bevacizumab, however, comparable efficacy was observed in treating macular edema secondary to CRVO, with significantly fewer intravitreal injections required with aflibercept [17]. Despite this, studies have found that aflibercept administration with a treat-and-extend dosing regimen was able to increase the macular edema-free interval in patients previously treated with ranibizumab or bevacizumab in patients with RVO [32,33,34]. Conbercept is a novel anti-VEGF-A, VEGF-B, and PLGF fusion protein with a similar structure to aflibercept and is approved for use in China [35]. Its safety, efficacy, and microvascular changes have been studied in smaller clinical trials [36]. Along with aflibercept, conbercept has been suggested to require fewer injections while being equally effective in improving visual acuity as compared to ranibizumab [37,38].

Other modalities used to treat RVO include laser photocoagulation and corticosteroids. Tachyphylaxis occurs when there is a diminished response to repeated anti-VEGF dosing, often a result of the upregulation of alternative angiogenic pathways and the persistence of the VEGF pathway [39]. Laser photocoagulation and corticosteroids remain useful adjuncts or alternatives when anti-VEGF therapy is suboptimal. In addition to suppressing VEGF, dexamethasone reduces other vasoactive and inflammatory proteins implicated in vascular permeability [40]. The safety and efficacy of repeated dexamethasone intravitreal implants have been demonstrated in patients with BRVO and CRVO-associated macular edema as compared to anti-VEGF injections, with the caveat that functional efficacy decreases and risk for intraocular pressure elevation and cataract progression increases over time with repeated injections [41,42,43,44,45,46,47,48,49]. Some studies have suggested that dexamethasone implantation can be used as an alternative in the case of refractory macular edema, with anti-VEGF treatment sometimes showing better functional and anatomical improvement [50,51,52,53,54,55].

### 2.3. Retinal Vein Occlusion—Emerging Therapies and Future Directions

Although intravitreal anti-VEGF has become a main component of RVO treatment, therapeutics that target both VEGF and the Ang-2/Tie pathways have gained interest [6]. Faricimab targets VEGF-A and Ang-2 (Table 1). A meta-analysis of studies comparing faricimab to anti-VEGF monotherapy found that faricimab resulted in similar improvements in BCVA but superior improvement in central retinal thickness (CST) compared to anti-VEGF monotherapy [56]. The BALATON and COMINO trials demonstrated sustained BCVA gains and CST reductions at long-term follow-up periods (24 and 72 weeks, respectively) with longer dosing intervals compared to aflibercept. This suggests increased durability of faricimab and the potential to reduce treatment burden compared to anti-VEGF monotherapy [57,58].

Lastly, biosimilars have been used for various ophthalmologic diseases, including RVO. Biosimilars are meant to mimic their reference products in terms of quality, safety, and efficacy, and are intended to be more cost-effective, thus potentially improving patient access and treatment adherence [56]. Ranibizumab biosimilar ranibizumab-nuna was the first biosimilar to receive FDA approval for the treatment of macular edema following RVO (as well as age-related macular degeneration and myopic choroidal neovascularization) [59]. Since then, ranibizumab-eqrn and aflibercept biosimilars (aflibercept-yszy, aflibercept-mrbb, aflibercept-jbvf, and aflibercept-ayyh) have received regulatory approval for the treatment of macular edema following RVO [60,61,62,63,64]. Bevacizumab biosimilars are not FDA-approved for the treatment of RVO. However, recent retrospective studies outside of the United States have shown effectiveness of bevacizumab BEVATAS and MVASI in India and Australia, respectively, in treating RVO [65,66].

Secondary neovascular glaucoma (NVG) requires aggressive treatment to prevent blindness and painful end-stage glaucoma. A common cause of NVG is iCRVO, which increases intraocular VEGF and stimulates blood vessel growth on the iris and the anterior chamber angle. Eventually, this growth can lead to the constriction of an irreversible fibrovascular membrane over the trabecular meshwork and iris, which ultimately obstructs aqueous humor outflow and causes increased intraocular pressure [67]. Aside from currently used anti-VEGF therapeutic agents, Aganirsen is a 25-mer phosphorothioate antisense oligonucleotide that inhibits new vessel formation by blocking IRS-1, an upstream marker in the VEGF and TNF-α cascade. The STRONG trial aimed to assess the efficacy and safety of aganirsen in preventing NVG in patients with iCRVO, focusing on the appearance of neovascularization and the rise in intraocular pressure after 24 weeks [68]. It currently has no analyses or follow-up data to date.

## 3. Age-Related Macular Degeneration

### 3.1. Age-Related Macular Degeneration—Classification and Pathogenesis

Age-related macular degeneration (AMD) is a leading cause of irreversible vision loss in the elderly population worldwide [69,70]. Clinically, advanced AMD is classified into two forms: neovascular “wet” AMD (exudative AMD or nAMD), and non-neovascular “dry” AMD (nonexudative AMD) [71]. Exudative AMD is driven by pathological choroidal neovascularization (CNV) from the choriocapillaris through Bruch’s membrane into the subretinal pigment epithelium or subretinal space. Another variant of nAMD is through retinal angiomatous proliferation (RAP) lesions, where abnormal intraretinal angiogenesis invades the subretinal space, resulting in the formation of retinal-choroidal anastomosis.

### 3.2. Age-Related Macular Degeneration—Current Anti-Angiogenic Therapies

Anti-VEGF therapy has revolutionized the management of nAMD, markedly improving visual outcomes compared to the prior use of photodynamic therapy [72]. Like RVO, agents such as ranibizumab, bevacizumab, and aflibercept are first-line therapies that are capable of inducing the regression of CNV, reducing macular fluid, and significantly lowering the risk of vision loss [73].

Brolucizumab is a single-chain antibody fragment that binds VEGF-A with high affinity. Due to its small size, at roughly 26 kDa, brolucizumab allows for a higher molar dose per injection and has the intended goal of achieving a longer duration of VEGF activity suppression. In phase III HAWK and HARRIER trials, brolucizumab administered every 12 weeks was non-inferior in visual outcomes compared to aflibercept every 8 weeks. However, a significant subset of patients (approximately 4–5%) developed severe intraocular inflammation, most notably retinal vasculitis and occlusive retinal vasculopathy, and thus brolucizumab is currently uncommonly used for nAMD.

In recent years, faricimab has been approved to extend dosing intervals further. In the TENAYA and LUCERNE trials, faricimab administered at intervals of up to 16 weeks maintained non-inferior vision gains compared to aflibercept every 8 weeks [73,74]. Approximately 80% of patients were able to be managed on 12-week dosing, and 45% managed on 16-week fixed dosing intervals [75]. Thus far, the ocular safety profile appears similar to aflibercept [74].

### 3.3. Age-Related Macular Degeneration—Emerging Therapies and Future Directions

Development of biosimilar anti-VEGF agents has also played a role in emerging treatment of nAMD. Ranibizumab-nuna and ranibizumab-eqrnm have also been approved for nAMD [76]. Ranibizumab-nuna has demonstrated equivalent improvement in visual acuity compared to reference ranibizumab at all time points throughout a 52-week period [76]. In 2023, aflibercept-jbvf and aflibercept-yszy were approved for nAMD treatment [77]. Finally, bevacizumab-vikg is currently under review by the FDA [77]. In the recent Norse Eight Trial, bevacizumab-vikg injections were non-inferior to ranibizumab in visual outcomes for nAMD [78].

Recent research in gene therapy for neovascular AMD centers on delivering viral vectors that include sustained expression of anti-angiogenic proteins to suppress VEGF-driven neovascularization and reduce injection burden. For example, RGX-314, delivered via subretinal or suprachoroidal injection, suppresses neovascularization via expression of a monoclonal antibody fragment like ranibizumab, which binds to VEGF-A (Table 2) [79]. EXG102-031 is another subretinal gene therapy candidate in Phase 1 testing [80]. Ixoberogene soroparvovex (ADVM-022) delivers an aflibercept coding sequence via an adeno-associated virus (AAV) capsid that is injected intravitreally rather than subretinally. 4D-150 is also an intravitreal gene therapy approach that has shown a favorable safety profile thus far in Phase 1/2 trials (Table 2) [81]. While anti-VEGF injections remain the cornerstone of neovascular AMD treatment, gene therapies could drastically reduce treatment burden by providing sustained anti-angiogenic effects [82]. The balance of long-term efficacy, safety, and access will determine how these emerging treatments will integrate into clinical practice.

Anti-complement drugs are now approved for treatment of geographic atrophy (GA), a severe form of AMD causing central vision loss. Pegcetacoplan, a complement C3 inhibitor, and avacincaptad pegol, a complement C5 inhibitor, are two agents with regulatory approval in the United States and are administered by intravitreal injection [83]. Phase 3 trials OAKS and DERBY demonstrated that pegcetacoplan can significantly slow the growth of GA lesions with an acceptable safety profile, however, with an increased dose-dependent risk of developing new-onset nAMD in treated eyes [84]. Avacincaptad pegol has demonstrated a significant reduction in GA lesion growth in the GATHER1 and GATHER2 phase 3 trials, with risk of macular neovascularization at slightly lower rates compared to pepgcetacoplan [85]. Other ongoing complement-targeted trials include pegcetacoplan (GALE trial), NAX007 (ARCHER trial), danicopan (ALXN2040), NGM621 (CATALINA trial), poxelimab and cemdisiran (SIENNA trial), and intravitreal BI-771716 (VERDANT trial) (Table 2) [86,87,88,89,90].

These are various ongoing clinical trials. High-dose aflibercept programs, including PULSAR in nAMD, are building on earlier phase 2 CANDELA findings [87]. Port delivery systems with ranibizumab are being evaluated in nAMD in the ARCHWAY and PORTAL trials [91,92]. Tyrosine-kinase inhibitor approaches include the EYP-1901 Phase 3 LUGANO trial in nAMD and two Phase 3 trials of OTX-TKI (axitinib implant), SOL-I, and SOL-R [93,94]. Gene therapy trials include RGX-314 (phase 2b/3 ATMOSPHERE), ADVM-022 (AAV.7m8-aflibercept in phase 2 LUNA study), and ixoberogene soroparvovex (Ixo-vec in phase 3 ARTEMIS study) (Table 2) [95,96,97].

## 4. Diabetic Retinopathy

### 4.1. Diabetic Retinopathy—Classifications and Pathogenesis

Diabetic retinopathy (DR) is a common microvascular complication of diabetes and a leading cause of vision loss in the working-age population [98]. DR is staged as non-proliferative diabetic retinopathy (NPDR) or proliferative diabetic retinopathy (PDR). Diabetic macular edema (DME) can occur in NPDR or PDR and is due to the breakdown of the BRB, increasing fluid influx into the neurosensory retina, and a decrease in drainage function by Muller glia and RPE cells, resulting in reduced fluid efflux out of the retina [98]. In response to significant ischemia that occurs in DR, increased hypoxia-induced angiogenic factors cause increased vascular permeability and prompt migration and proliferation of endothelial cells, leading to neovascularization in PDR [99]. The concentration of hypoxia-induced VEGF in ocular fluids correlates with the breakdown of the BRB and disease severity in DR [100]. Along with other growth factors and cytokines, Ang-2 is markedly elevated in the vitreous of patients with PDR and DME [101].

### 4.2. Diabetic Retinopathy—Current Anti-Angiogenic Therapies

Given the central role of VEGF, anti-VEGF therapies have become a keystone in the management of DR, particularly as it relates to DME and PDR (Table 1 and Table 2). The standard of care for sight-threatening DME now involves intravitreal anti-VEGF agents, which can reduce macular thickening and improve visual acuity in a significant fraction of patients [102]. In multiple large trials (Protocol S, VIVID/VISTA, RISE/RIDE), intravitreal anti-VEGF agents led to superior vision outcomes in DME compared to previous standards such as focal laser [102,103]. Based on landmark studies such as the Protocol S trial, which compares ranibizumab versus laser therapy for PDR, anti-VEGF therapy has been shown to be as effective as pan-retinal laser photocoagulation in the short-to-medium term [104]. A meta-analysis indicates that anti-VEGF therapy in PDR can additionally yield a slight mean visual acuity gain (+2 to +3 letters), whereas PRP on average preserves vision loss (+0 to +1 letter) [105]. Moreover, anti-VEGF reduces the risk of vitreous hemorrhage and retinal detachment in proliferative disease compared to observation. For these reasons, anti-VEGF injections are increasingly used either as primary therapy for PDR or combined with PRP [106].

Despite the efficacy of anti-VEGF agents, there are multiple limitations to this therapy in the context of DR. Many DME patients require frequent injections over the years to maintain edema control. Furthermore, some patients may exhibit a sub-optimal response [107]. Real-world studies show that visual gains in DME are less than those observed in clinical trials, likely due to under-treatment [108]. Furthermore, anti-VEGF monotherapy primarily addresses vascular leakage and proliferation but does not reverse underlying diabetic ischemia and retinal neurodegeneration [98]. For peripheral retinal ischemia, laser photocoagulation is a useful tool. Destroying hypoxic retinal tissue that produces VEGF reduces the metabolic drive for neovascularization.

Other adjunctive therapies, especially with DME, include intravitreal corticosteroids, such as dexamethasone or a fluocinolone implant. Steroids are particularly considered in DME patients who are pseudophakic or cannot maintain monthly visit schedules [109]. Combined therapy with dexamethasone and anti-VEGF therapy has shown a greater reduction in central subfield thickness compared to anti-VEGF monotherapy in patients with persistent DME [110].

### 4.3. Diabetic Retinopathy—Emerging Therapies and Future Directions

In DR, faricimab plays a key role in targeting both VEGF-A and Ang-2. Faricimab demonstrated non-inferior vision gains to aflibercept, with the potential to extend dosing intervals [111]. The phase III YOSEMITE and RHINE trials evaluated faricimab in DME using personalized dosing up to Q16 weeks. At one year, faricimab at adjusted intervals achieved similar visual improvement as fixed Q8 weeks aflibercept, with approximately 52% of patients being maintained on >12-week dosing by week 52 [111]. Another investigation avenue is combination therapy targeting multiple angiogenic pathways. The co-formulation of aflibercept and anti-Ang-2 antibodies, such as nesvacumab, was studied in the phase 2 RUBY and ONYX trials. However, these studies did not show a significant gain over aflibercept alone [112].

Like in RVO and AMD, anti-VEGF biosimilars are being used for the treatment of DME. The INSIGHT trial demonstrated that aflibercept biosimilar MYL-1701P is equivalent in efficacy and safety to reference aflibercept [113]. Furthermore, bevacizumab biosimilars have been studied as an off-label treatment for DME in India [114]. Similar to RVO, four aflibercept biosimilars, aflibercept-yszy, aflibercept-mrbb, aflibercept-jbyf, and aflibercept-ayyh, and ranibizumab-eqrn have been FDA-approved for DME [59,60,61,62,63,64].

## 5. Comparison of Mechanism, Therapeutic, and Translational Implications Between RVO, AMD, and DR

RVO, AMD, and DR share a final common pathway of VEGF-driven vascular permeability and neovascularization, and, as a result, anti-VEGF injections remain the primary treatment modality across all three disease processes. However, they arise from distinct upstream mechanisms. In RVO, an acute venous outflow obstruction leads to ischemia. In nAMD, a chronic degenerative process involving RPE dysfunction underlies choroidal neovascularization. In DR, diffuse microvascular injury from hyperglycemia and low-grade inflammation leads to capillary dropout, breakdown of the blood-retina barrier, and proliferative DR. Translationally, these differences provide indication-specific strategies, such as ischemia-guided PRP in DR and iRVO, and complement cascade targeted therapy in geographic atrophy.

In addition, DME has higher inflammatory markers than RVO or AMD, with consistently higher intraocular levels of inflammatory cytokines IL-6 and MCP-1 than BRVO [115]. In AMD, inflammation seems to be more a consequence of chronic retinal pigment epithelium and Bruch’s membrane alterations than primary diffuse retinal inflammation [7]. This distinction explains the higher demonstrated efficacy of corticosteroids in DME as compared to RVO and AMD [110]. Corticosteroids have not demonstrated significant benefit in randomized trials in AMD, and their use is limited to refractory cases, often in combination with VEGF [116,117]. In RVO, corticosteroids reduce macular edema and improve vision, but anti-VEGF agents have been demonstrated to be superior to steroids for both visual and anatomical outcomes [50,51,52,53,54,55]. There is merit to combined steroid and anti-VEGF therapy, lengthening the average time to anti-VEGF reinjection in retinal diseases such as RVO [118]. However, there are mixed results on this, particularly in studying whether anti-VEGF and triamcinolone acetonide combination therapy requires fewer injections in patients undergoing macular edema [119,120,121]. This is in contrast to the well-established literature on the effectiveness of targeting inflammatory pathways with intravitreal corticosteroids in patients with DME [7].

## 6. Retinopathy of Prematurity

### 6.1. Retinopathy of Prematurity—Classifications and Pathogenesis

Retinopathy of prematurity (ROP), primarily observed in preterm infants, is characterized by aberrant neovascularization within avascular regions of the retina. It has the potential for detrimental outcomes, such as retinal detachment and blindness. It is the leading cause of preventable childhood blindness globally. Pathophysiology involves two postnatal phases. The first phase is characterized by hyperoxia exposure, followed by the loss of oxygen-regulated angiogenic growth factors and nutrients, and the provision of growth factors at the maternal–fetal interface. This results in attenuated retinal vessel growth and loss of some existing retinal vessels. In the second phase, retinal hypoxia in the poorly vascularized retina leads to increased concentrations of oxygen-regulated factors, such as EPO and VEGF, due to elevated levels of IGF-1. Newly formed vessels that poorly perfuse the retina are very permeable, causing fibrous scar formation and retinal detachment [122]. Interestingly, the association between systemic circulating VEGF levels and ROP severity in infants is inconsistent [123].

According to the Committee for Classification of ROP, the extent and severity of ROP are described in terms of location, severity, extent, and vascular dilatation and tortuosity (plus disease) in one of the three zones of the retina (zones I–III) [124]. In 2005, the classification of ROP was updated to include aggressive posterior retinopathy of prematurity (APROP) [125]. Previous treatment strategies for infants with type 1 ROP involve peripheral retinal ablation by cryotherapy. This technique has largely been replaced by laser therapy and, in the past decade, anti-VEGF therapy.

### 6.2. Retinopathy of Prematurity—Current Anti-Angiogenic Therapies

Current agents that have been studied for use in ROP include bevacizumab, ranibizumab, aflibercept, conbercept, and pegaptanib (a pegylated anti-VEGF aptamer) [126,127,128,129]. By targeting VEGF, these drugs can suppress angiogenesis and reduce the downstream effects of neovascular proliferation. Bevacizumab and ranibizumab are the two most common anti-VEGF drugs used for ROP [130,131]. Studies comparing the two therapies have found efficacy in controlling acute ROP, with limited systemic exposure to ranibizumab [132,133]. Accordingly, laser therapy was associated with a lower rate of recurrence compared to ranibizumab, particularly in type 1 ROP (Zones I and II combined). Aflibercept and bevacizumab demonstrated a longer duration of action and lower rates of recurrence [134,135]. The BUTTERFLEYE and FIREFLEYE trials found no clinically meaningful difference in favor of laser compared to aflibercept, with limited systemic drug exposure [136,137]. Combination therapy with both anti-VEGF and laser for ROP has been evaluated in a few studies, which suggest that combination therapy may decrease the rate of ROP recurrence, as demonstrated in anti-VEGF monotherapy, with later use of laser therapy post-anti-VEGF injection associated with less myopic outcomes [138,139,140]. There are concerns regarding the possible escape of intravitreal anti-VEGF agents into systemic circulation and their effect on organogenesis. A systematic review and meta-analysis found no systemic complications directly attributed to anti-VEGF treatment in 585 infants [141]. Multiple groups have investigated neurodevelopmental outcomes following anti-VEGF intravitreal administration [142]. Studies that have found more debilitating neurodevelopmental changes in infants managed by bevacizumab compared to laser therapy were retrospective and nonrandomized, with infants in the intravitreal injection group having more systemic issues. Overall, anti-VEGF agents used in treating ROP appear to be effective with a favorable safety profile; however, further randomized clinical trials are required to determine the preferred treatment, optimal dosing, and long-term ocular and systemic outcomes following treatment.

## 7. Conclusions

In this review, we have discussed normal angiogenesis and contrasted pathological angiogenesis in four diseases: retinal vein occlusion, age-related macular degeneration, diabetic retinopathy, and retinopathy of prematurity. It is critical to understand the molecular pathways involved with pathological angiogenesis to appreciate how current and emerging therapies leverage its effects. Although anti-angiogenesis drugs such as VEGF-targeting therapies have become mainstay treatments for retinal diseases, there remain further challenges with implementing anti-angiogenic treatments. Studies examining patients’ adherence to their prescribed treatment regimens have found non-adherence rates ranging from 17.5% to 30% [143]. Long-term and frequent intravitreal therapy imposes substantial socioeconomic and accessibility burdens, which lead to undertreatment [144,145]. Current preclinical research aims to generate longer-acting modalities, such as sustained-release implants, as well as alternative targets (Table 2) [146]. The purpose of these methods is to reduce the frequency of injections, improve adherence to anti-angiogenic regimens, and enhance the prognosis for visual acuity in various retinal diseases. Lastly, non-anti-VEGF therapies that may include inhibitors of pro-angiogenic factors such as PDGF are under preclinical investigation. They are designed to overcome the limitations of anti-VEGF monotherapy, such as tachyphylaxis and incomplete responses [147]. The need for non-anti-VEGF therapies stems from the multifactorial nature of retinal neovascularization, which may not be reflected adequately in animal models for human retinal angiogenesis [148]. Further research on the molecular pathways of angiogenesis in both animal models and human-relevant systems can help elucidate novel therapies and improve clinical translation in these vision-threatening conditions.

## Figures and Tables

**Table 1 cells-14-01849-t001:** Role of angiogenesis in retinal diseases and advances in drug development.

Target	Role in Retinal Disease	Drug Class	Key Examples	Clinical Status
VEGF/VEGFR	Central mediator of pathological angiogenesis in RVO, AMD, DME, ROP	Monoclonal antibodies, fusion proteins, aptamers, biosimilars, gene therapy	Ranibizumab, Aflibercept, Bevacizumab, Conbercept, Pegaptanib, VEGFR TKIs	First-line therapy
Angiopoietin/Tie2	Synergistic with VEGF	Bispecific antibodies	Faricimab (VEGF-A/Ang-2 bispecific)	Faricimab approved; others in clinical trials
Inflammatory Pathways	Inflammation (via inflammatory mediators) contributes to neovascularization and disease progression	Steroids, small molecules, biologics targeting inflammatory mediators	Triamcinolone, Dexamethasone	Off-label Early clinical/preclinical development

**Table 2 cells-14-01849-t002:** Novel clinical trials and their major anti-angiogenic drugs.

Drug	Clinical Trial	Retinal Disease
Ranibizumab	DIAMOND (eye drops) (Phase 3)	DME
	ARCHWAY (Port Delivery System for Continuous Delivery) (Phase 3)	Neovascular AMD
	PAGODA (Port Delivery System) (Phase 3)	DME
	PORTAL (Port Delivery)	Neovascular AMD
Aflibercept	QUASAR (8 mg) (Phase 3)	CME in RVO
	ELARA (8 mg) (Phase 3b)	Neovascular AMD and DME
	CANDELA (high dose) (Phase 2)	Neovascular AMD
	PHOTON (high dose)	DME
	PULSAR (high dose) (Phase 3)	Neovascular AMD
ADVM-022 (AAV.7m8-aflibercept)	LUNA (Phase 2)	Neovascular AMD
RGX-314	RGX-314 (Phase 1/2a)	nAMD (investigational)
Tarcocimab Tedromer	GLOW2 (Phase 3)	Moderately Severe to Severe NPDR
	DAYBREAK (Phase 3)	Neovascular AMD
Intravitreal OTX-TKI (axitinib implant)	SOL-I (Phase 3)	Neovascular AMD
	Sol-R (Phase 3)	Neovascular AMD
Intravitreal BI 764524	CRIMSON	Moderately Severe to Severe NPDR
ONS-5010 Compared to Lucentis	NORSE EIGHT	Neovascular AMD
EYE103	BRUNELLO (Phase 2/3)	DME
	BAROLO (Phase 2/3)	DME
EYP-1901, a Tyrosine Kinase Inhibitor (TKI)	LUGANO (Phase 3)	Neovascular AMD
OLN324	Phase 1b	Neovascular AMD, DME
Ixoberogene soroparvovec (Ixo-vec)	ARTEMIS (Phase 3)	Neovascular AMD
AR-14034 Sustained Release Implant	NOVA-1 (Phase 1/2)	Neovascular AMD
Ro6867461	RHINE (Phase 3)	DME
ABP 938	Phase 3	Neovascular AMD
Intravitreal KSI-301	BEACON (Phase 3)	CME RVO
	GLIMMER (Phase 3)	DME
	DAYLIGHT	Neovascular AMD
	GLOW (Phase 3)	Moderately Severe to Severe NPDR
SOK583A1	SANDOZ	Neovascular AMD
RGX314 Gene Therapy	ATMOSPHERE (Phase 2b/3)	Neovascular AMD

## Data Availability

The data presented in this study are available in PubMed at [https://pubmed.ncbi.nlm.nih.gov (accessed on 9 November 2025)]. Please refer to the reference list below for more details.

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
