# Peer review of "Role of Angiogenesis in Retinal Diseases and New Advances in Drug Development"

_cells, 2025, doi:10.3390/cells14231849_

Round 1
Reviewer 1 Report (Previous Reviewer 2)
Comments and Suggestions for Authors
Please keep the same tone/narrative throughout the manuscript. In some parts the paper reads more like a textbook style manuscript, whereas in other parts it is like literature enumeration.
It should be more ovious that you emphasize on the mechanistic integration and critical evaluation of the information included. E.g., in "Current anti-angiogenic therapies for RVO" or "Current Anti-Angiogenic Therapies for AMD", these sections are overly descriptive, providing lists of available drugs and trials with no evidence of analytical synthesis. Reduce redundant or excessive pharmacological detail and emphasize on comparative mechanisms, therapeutic paradigms, limitations, and translational implications. Consider including figures to illustrate these mechanisms.
The introduction is thorough but quite long with too many references about developmental biology that could be reduced. Although you establish what will follow, you dedicate too much of your manuscript on embryological choroidal and retinal development, which is actually not directly relevant to the focus of your review. Reduce the developmental backround to a smaller paragraph summarizing main key points, such as ascular layers, hypoxia-driven VEGF signaling. Underline the he rationale for anti-angiogenic therapy in ocular pathology.
The review may benefit from a more thorough discussion of therapeutic challenges:
- Mechanisms of anti-VEGF resistance or tachyphylaxis
- The role of inflammation and fibrosis in chronic neovascular diseases
- Limitations of current animal models for human retinal angiogenesis
- Socioeconomic and accessibility implications of long-term intravitreal therapy
I believe that the review need 1-2 schematic figures summarizing:
- Physiologic vs. pathologic angiogenesis in the retina/choroid.
- Mechanistic overview of VEGF, Ang/Tie
Comments on the Quality of English Language
Frequent minor grammatical and syntactic issues; please revise the whole manuscript accordingly.
Author Response
Thank you for the comments and suggestions. We have revised the manuscript to maintain a consistent tone/narrative throughout. We have also improved minor grammatical and syntactic issues.
Question 1: Please keep the same tone/narrative throughout the manuscript. In some parts the paper reads more like a textbook style manuscript, whereas in other parts it is like literature enumeration.
Answer 1: Thank you for your comments. We think that inclusion of classifications and the discussion of pathogenesis in these sections is crucial to give more context to the disease process and anti-angiogenic therapies. However, we have updated the beginning sections of each disease (2.1, 3.1, 4.1, 5.1) to minimize the amount of “textbook style” text in this manuscript and avoid repetitive use of hypoxia-related factors that was originally mentioned and explained in the introduction. Please see lines 89-94, 195-203, 271-282, and 360-379 to see these changes.
Question 2: It should be more obvious that you emphasize on the mechanistic integration and critical evaluation of the information included. E.g., in "Current anti-angiogenic therapies for RVO" or "Current Anti-Angiogenic Therapies for AMD", these sections are overly descriptive, providing lists of available drugs and trials with no evidence of analytical synthesis. Reduce redundant or excessive pharmacological detail and emphasize on comparative mechanisms, therapeutic paradigms, limitations, and translational implications. Consider including figures to illustrate these mechanisms.
Answer 2: Thank you for your comments, we have cut out portions of the RVO, AMD, and DR sections and added another paragraph to the manuscript that compares the mechanism of DME, RVO, and AMD and therapeutic implications. Please see lines 335-357 for these changes.
Question 3: The introduction is thorough but quite long with too many references about developmental biology that could be reduced. Although you establish what will follow, you dedicate too much of your manuscript on embryological choroidal and retinal development, which is actually not directly relevant to the focus of your review. Reduce the developmental background to a smaller paragraph summarizing main key points, such as vascular layers, hypoxia-driven VEGF signaling. Underline the rationale for anti-angiogenic therapy in ocular pathology.
Answer 3: Thank you for your comments. We have cut down the developmental background of the introduction section and re-arranged the structure of the section so that the progression of angiogenesis in retinal vascular development is easier to follow. By doing this, it also makes more clear the rationale for anti-angiogenic therapy in ocular pathology. Please see lines 40-49 for these modifications.
Question 4:
The review may benefit from a more thorough discussion of therapeutic challenges:
- Mechanisms of anti-VEGF resistance or tachyphylaxis
- The role of inflammation and fibrosis in chronic neovascular diseases
- Limitations of current animal models for human retinal angiogenesis
- Socioeconomic and accessibility implications of long-term intravitreal therapy
Answer 4: Thank you for your suggestions. We have modified the manuscript to reflect your suggestions.
- Please see lines 156-159 to see that we incorporated a discussion of tachyphylaxis into the use of corticosteroids and laser photocoagulation treatment for RVO
- Please see lines 66-76 to see how we added the role of inflammation and fibrosis in chronic neovascular diseases.
- Please see lines 423-430 to see how we incorporated the limitations of current animal models for human retinal angiogenesis into our conclusion, linking it with limitations in creating non-anti-VEGF therapies that are necessitated due to the multifactorial nature of retinal neovascularization.
- Please see lines 417-419 to see how we incorporated the socioeconomic and accessibility implications into our conclusion when we also discuss preclinical research focusing on longer-acting modalities.
Question 5: I believe that the review need 1-2 schematic figures summarizing
- Physiologic vs. pathologic angiogenesis in the retina/choroid.
- Mechanistic overview of VEGF, Ang/Tie
Answer 5: Thank you for your comments. There are multiple published figures that review physiologic vs. Pathologic angiogenesis in the retina/choroid and the mechanistic overview of VEGF and Ang/Tie. We have referenced papers that contain these figures in this manuscript. Please see References 4, 6, 13 for this.
Question 6: Frequent minor grammatical and syntactic issues; please revise the whole manuscript accordingly.
Answer 6: Thank you for your comment. We have improved the spelling and grammar mistakes. Please see lines 127 and 148 for examples of small modification that we have made to ensure proper spelling and grammar.
Reviewer 2 Report (Previous Reviewer 1)
Comments and Suggestions for Authors
The revised manuscript is improved. However, there are still many problems with grammar and spelling that must be corrected.
Comments on the Quality of English LanguageLanguage is OK, but there are many spelling and grammar errors as well as inappropriate hyphenation.
Author Response
Thank you for the comments and suggestions. We have modified the manuscript so that the grammar and spelling mistakes have been corrected.
Question 1: The revised manuscript is improved. However, there are still many problems with grammar and spelling that must be corrected.
Comments on the Quality of English Language: Language is OK, but there are many spelling and grammar errors as well as inappropriate hyphenation.
Answer : Thank you for your comment. We have improved the spelling and grammar mistakes. Please see lines 127 and 148 for examples of small modification that we have made to ensure proper spelling and grammar.
Reviewer 3 Report (New Reviewer)
Comments and Suggestions for Authors
- line 134: there is no space between studies and comparing; please correct;
- line 153: spelling error - cto instead of to; please correct;
- line 162: can can; please delete one can
- AMD section, 3.3. subchapter: please update this section with the new class of anti-complement drugs for GA;
- DR section, 4.2. subchapter:: steroids and anti-VEGF can "work" together in PDR with DME; please update this section with some studies;
- biosimilar molecules are not only indicated in AMD, but also in DME and ocular NV following iCRVO; please update.
Author Response
Thank you for your comments and suggestions. We have modified that manuscript to address specific spelling and grammar concerns. We have also updated disease sections to reflect added therapies.
Question 1:
- line 134: there is no space between studies and comparing; please correct;
- line 153: spelling error - cto instead of to; please correct;
- line 162: can can; please delete one can
Answer 1: Please see lines 127 and 148 for the appropriate changes.
Question 2: AMD section, 3.3. subchapter: please update this section with the new class of anti-complement drugs for GA.
Answer 2: Thank you for the comment. We have updated the manuscript to add anti-complement drugs for GA secondary to section 3.3 of AMD. Please see lines 247-256..
Question 3: DR section, 4.2. subchapter:: steroids and anti-VEGF can "work" together in PDR with DME; please update this section with some studies;
Answer 3: Thank you for your comment. We have updated Section 4.2 to include citations that demonstrate the use of steroids in DME cases. Please see lines 311-313.
Question 4: biosimilar molecules are not only indicated in AMD, but also in DME and ocular NV following iCRVO; please update.
Answer 4: Thank you for your comment. Please see lines 172-180 and 327-332 to see the added biosimilar information to the RVO and DR sections.
Round 2
Reviewer 1 Report (Previous Reviewer 2)
Comments and Suggestions for Authors
Thank you for addressing the suggested comments. I have no further concerns regarding your manuscript.
Comments on the Quality of English LanguageNo further issues
This manuscript is a resubmission of an earlier submission. The following is a list of the peer review reports and author responses from that submission.
Round 1
Reviewer 1 Report
Comments and Suggestions for Authors
This manuscript reviews research articles on clinical trials of anti-angiogenic therapies in the last 20 years. The focus is on therapies that target VEGF and its receptor VEGFR, the angiopoietin/Tie1 pathway, and inflammatory pathways. There are a number of problems with the presentation . Numerous spelling and English grammar mistakes detract from the presentation and the text is often confusing. The authors should seek profession assistance to correct the numerous mistakes and improve clarity. There are too many spelling errors, grammatical mistakes, incomplete sentences, and confusing statements throughout the text to list here, but a few examples from the first few pages are outlined below.
- Lines 19-20. The abstract states that four diseases are discussed, but only 3 are listed. What about retinopathy of prematurity?
- Line 40. Relie is not a word.
- Line 45. What is hemo-vasculogenesis?
- Lines 49-51. What are Sattler’s and Haller’s layers? What is CD30+? What is Ki67 and why is proliferation of Ki67+ cells important? What is NG2?
- Line 103. What is the meaning of “abnormal angiogenesis development of macular edema”?
- Lines 119 and 121. What is the meaning of “non-inferior”?
- Lines 143-144. What is the meaning of “reduced intraviteal injection burden”?
- The format of reference citations in the text should be corrected to place the bracketed reference numbers before the punctuation, not after it.
The authors should seek professional assistance to improve English language and clarity
Reviewer 2 Report
Comments and Suggestions for Authors
This paper is a comprehensive narrative review on the role of angiogenesis in retinal diseases summarizing approved and emerging anti-angiogenic therapies. The manuscript is clear, well-structured and nicely written. However, I feel that it has rather limited novelty because it presents known information rather than new insights or perspectives. There are similar reports in various journals over the last 5 years.